# Virtual Restoration and Visualization Changes through Light: A Review

**Ángela Gómez Manzanares [1],*, Anto J. Benítez [2]** and **Juan Carlos Martínez Antón [1]**

1   Department of Optics, University Complutense of Madrid, 28040 Madrid, Spain; jcmartin@ucm.es
2   Department of Communication. University Carlos III of Madrid, 28903 Madrid, Spain; abenitez@hum.uc3m.es
*   Correspondence: anggomez@ucm.es

**Abstract:** The virtual modification of the appearance of an object using lighting technologies has become very important in recent years, since the projection of light on an object allows us to alter its appearance in a virtual and reversible way. Considering the limitation of non-contact when analysing a work of art, these optical techniques have been used in fields of restoration of cultural heritage, allowing us to visualize the work as it was conceived by its author, after a process of acquisition and treatment of the image. Furthermore, the technique of altering the appearance of objects through the projection of light has been used in projects with artistic or even educational purposes. This review has treated the main studies of light projection as a technique to alter the appearance of objects, emphasizing the calibration methods used in each study, taking into account the importance of a correct calibration between devices to carry out this technology. In addition, since the described technique consists of projecting light, and one of the applications is related to cultural heritage, those studies that carry out the design and optimization of lighting systems will be described for a correct appreciation of the works of art, without altering its state of conservation.

**Keywords:** projection mapping; calibration; lighting; cultural heritage

## 1. Introduction

The colour and appearance of an object depend on the physical and chemical properties of the object itself, the source of visible electromagnetic energy that illuminates it, and the observer who detects the energy reflected by the object [1]. Taking into account these three factors, it is possible to think that by projecting adequate lighting on an object, its appearance could be altered.

From this idea, the term Mixed Reality arises, in order to alter the perception of reality including virtual content. Milgram and Kishino defined the concept of mixed reality as a subset of the technologies related to virtual reality, which includes Augmented Reality (AR) and Augmented Virtuality (AV). Thus, they defined Milgram's Reality–Virtuality Continuum (Figure 1), which is a line on which the virtual space is located at one end, real space at the other, and AR and AV in the center. AR was defined by Milgram and Kishino as the technology that improves the visualization of a real environment through virtual objects and AV was defined as the opposite of AR. Therefore, in Figure 1, AR is closer to reality and AV closer to virtuality [2].

Taking these factors into account, it is possible to think that by projecting adequate lighting on an object, its appearance could be altered in a virtual and therefore reversible way. Thereby, the technique of Spatial Augmented Reality (SAR) or projection mapping arises, which uses optical elements and video projectors, holograms, radio frequency labels and other tracking technologies to display virtual information directly on an object altering their appearance [3–5].

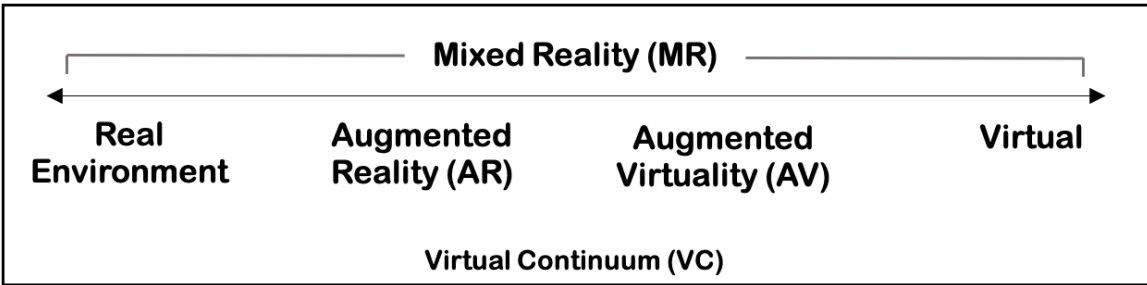

**Figure 1.** Milgram's Reality–Virtuality Continuum [2].

Azuma, in 1997, completed the definition of Milgram and Kishino, defining AR as 'a technology that allows the user to see the real world, with virtual objects superimposed or compounded with the real world' [6].

In recent years, AR has become more important in fields such as education, entertainment, conservation, restoration and adequate exhibition of cultural heritage, among others [3,7].

The projection mapping technique uses cameras to capture the object to be illuminated and projectors to project the desired image onto the object. It has the advantage over other AR techniques that it doesn't require viewers to use any type of glasses or instrument to appreciate the effect, allowing multiple users to appreciate the effects of projection mapping technology at the same time [5].

Section 2 of this article collects those studies which use the projection mapping technique in order to alter the appearance of an object, with special emphasis on those studies that use the technique as a method of displaying cultural heritage.

For the development of the projection mapping technique, a meticulous calibration method is required that allows adequate synchronization between the instruments used. Then, it is necessary that the image be correctly located on the object, avoiding unwanted effects. Section 3 of this article lists the main calibration techniques used in projection mapping in recent years.

The complexity of implementing this technique in the field of conservation of cultural heritage objects consists in the development of a lighting system that allows a safe display of the artwork without producing alterations in its appearance with respect to when it was made by the artist. This is due to the photochemical and thermal effect of light, causing degradation of the materials it illuminates. Section 4 of this article collects the most significant techniques to address this difficulty.

## 2. Projection Mapping Applications

The ability of the projection mapping technique to alter the appearance of an object by projecting light on it has made its applications more and more common every day, as shown in Figure 2. Under the name of Shader Lamps, Raskar in 2001 introduced the idea of animating white objects without texture by projecting different graphic images on them, altering the appearance of these objects by varying their perception of colour and texture [8].

Amano, by means of superimposed projection, altered the appearance of contrast in real objects that already had their own colourimetric and texture properties [9].

In Wang et al. (2010), the concept of context-aware light source was introduced, defined by them as 'a light source that can modify its lighting depending on the sensor information obtained from the scene'. It is a projection mapping application, which works by means of the previous acquisition of the information of the scene by a camera, so that after the appropriate image treatment, the improvements are shown on the object through a projector. In this way, it allows improving the appearance of an object in real time, even if the user manipulates it. From a context-aware light source, they developed the concept of proxy light, designing a portable instrument which, as if it were a common flashlight, allows the user to illuminate the areas of the object they want, appreciating parts of the object that are not perceptible to the naked eye by humans. This instrument is a very valuable tool for museumgoers, archaeologists, and art restorers [10].

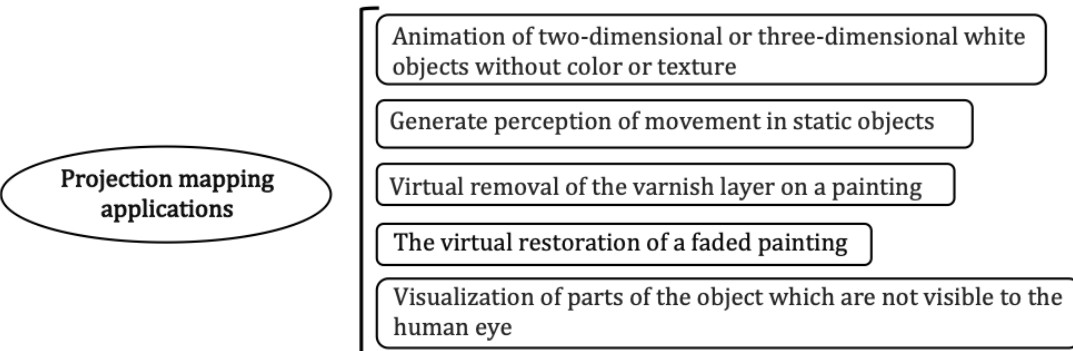

**Figure 2.** Outstanding applications of the projection mapping technique.

Later, in 2014, Revealing Flashlight emerged. It is an instrument that, like the one described by Wang et al., allows users to visualize hidden parts of an object. Wang et al.'s instrument works in real time, so it is limited by the camera resolution and 2D image processing, Revealing Flashlight performs 3D object pre-analysis, then the displayed images are not limited by the camera resolution or by analysis time, thus allowing a more detailed visualization of the object, obtaining the results shown in Figure 3 [11].

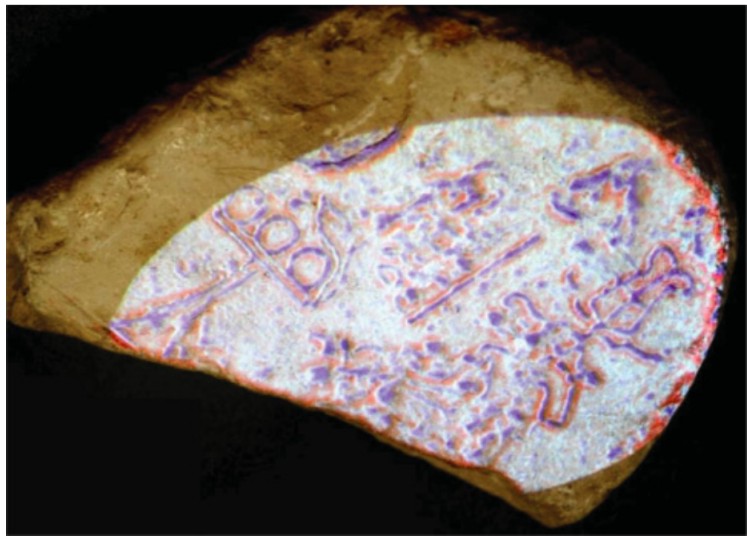

**Figure 3.** Image proposed by Ridel et al. [11] where the result is shown after illuminating a cultural artifact with the Revealing Flashlight technique.

Both Punpongsanon et al. and Kawabe et al. used the projection mapping technique in combination with psychophysical studies to alter the perception of movement through lighting [12,13]. Kawabe et al. developed a technique called Deformation Lamps to achieve the perception of the movement of objects while maintaining their original colour and texture [12]. Punpongsanon et al. used the mapping projection technique to alter the perception of bending stiffness of a fabric [13].

Among the applications mentioned above, the application related to the conservation and exhibition of cultural heritage objects will be described in more depth.

Some artworks have had their appearance affected due to the action of external agents, such as non-optimal humidity/temperature conditions or the incidence of electromagnetic radiation from the sun or different sources of artificial light over their years of exposure [14–16]. A restoration by a classical method can then be complicated. However, by means of the projection mapping technique, a restored vision of the artwork can be obtained by projecting an image on it, called in many studies the compensation image.

Lafontaine presented the precursor of this technique in 1986: he developed a projection technique by which an artwork can be visualized without affecting the yellowing that occurs in its varnish with the incidence of electromagnetic radiation on it [17].

The projection mapping technique was used in 2005 by Peral to virtually restore the appearance of the portico of Saint Mary's Cathedral in Vitoria (Spain) [18].

Aliaga et al. (2008) presented a technique to restore deteriorated objects by projecting light with multiple overlapping projectors. They proposed an interactive restoration algorithm, in which users could select those points they want to restore in a captured image. Then, using a restoration algorithm, one could obtain the compensation image and thus project it onto the object [19].

Aleksić and Jovanović used the virtual restoration technique through projection mapping to obtain a restored view of Lazar Vozarević's *Untitled* from 1961. To do this, they were inspired by the technique developed by Stenger in restoring Mark Rothko's murals, since both artists used very similar chemical compositions for the development of their work. But unlike Stenger's work, Vozarevic's work has a three-dimensional geometry, requiring a different restoration technique [20].

Recently, Vázquez et al. (2020) presented a method of restoring artworks by projecting light point by point, characterized by the previously acquired spectral reflectance of the artwork. For their procedure, they used a photograph of Sorolla's *Walk on the Beach* that was previously artificially aged using a Matlab algorithm, as shown in Figure 4 [21].

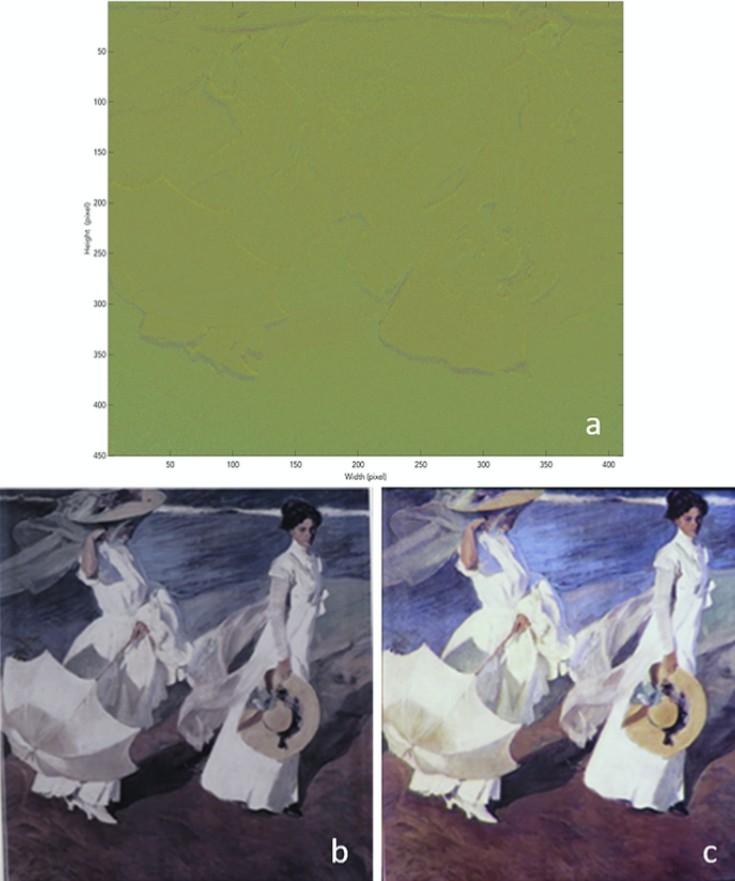

**Figure 4.** Images proposed by Vázquez et al. [21] where it is shown: (**a**) RGB color matrix that will be projected onto the aged image to compensate for the aged appearance of the artwork. (**b**) Photograph of Sorolla's *Walk on the Beach* artificially aged. (**c**) Photograph of Sorolla's *Walk on the Beach* after virtual restoration process.

## 3. Calibration

Projection mapping systems use at least one camera and one projector, so that the camera can detect and adjust the image from the projector [5]. In order for an image projected onto the real object to be properly visualized, a series of geometric and photometric calibration algorithms are required [5]. This section shows the evolution of the calibration algorithms designed for projection mapping projects (Figure 5).

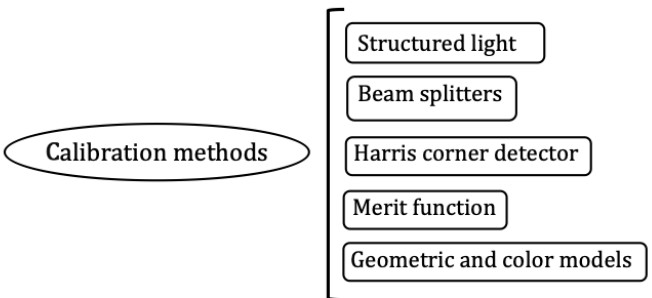

**Figure 5.** Calibration methods used in the studies shown in this review article.

The technique of Aliaga et al. initially consisted of acquiring a geometric model of the object to be restored and the proper calibration of the projectors based on the self-calibration method they described in [19,22]. Subsequently, the image of the object was captured and restored using an interactive energy minimization algorithm. Finally, the projection image that would be projected onto the object was calculated [19].

They obtained a self-calibrated structured light method allowing data processing from multiple points of view, being able to obtain a 3D object reconstruction. For the correspondence between the pixels they exploited the duality of cameras and projectors [22].

For the development of Revealing Flashlight, Ridel et al. used the calibration method that Audet et al. had proposed in 2010 [11,23]. Audet et al. had developed an alignment algorithm between camera, object and projector [23]. In their method, they described two models, a geometric model and a colour model, through which the necessary information can be obtained to predict how the image projected on the camera sensor is formed. They were based on the pinhole camera model to obtain Equations (1) and (2), which define the projection in the image plane of a camera placed at the origin and a calibrated projector, respectively, for a point $x_3$ located in the plane of the surface [23].

$$x_c = K_c(Ix_3 + 0), \tag{1}$$

$$x_p = K_p(R_px_3 + t_p), \tag{2}$$

where $K$ is the camera matrix, which contains the internal or intrinsic projective parameters, and where $R$ and $t$ are parameters that shape the orientation and position of the devices [23].

They developed a colour model so that the system could be easily calibrated without camera control with a single projector and flat surface (Figure 6). For this, they formulated Equation (3), which allows predicting the colour information that the camera will observe ($p_c$) knowing the colour emitted by the projector ($p_p$) and the reflectance emitted on the surface ($p_s$).

$$p_c = p_s[gX_{3x3}p_p + a] + b, \tag{3}$$

where $g$ is the gain of the projector light; $X$ is the colour mixing matrix; $a$, the ambient light; and $b$, the noise bias of the camera. All vectors are three-vectors in the RGB colour space [23].

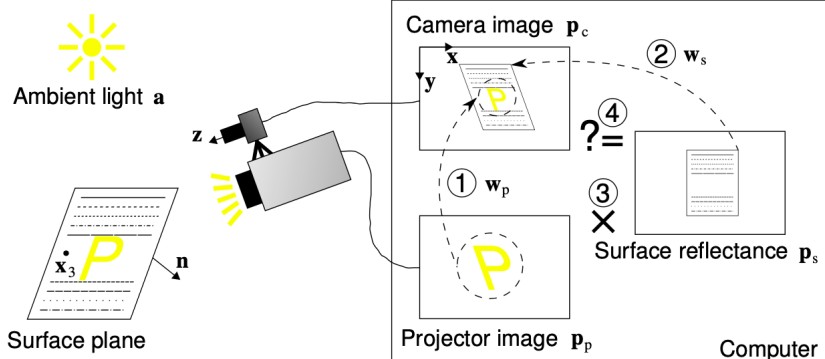

**Figure 6.** Sketch proposed by Audet et al. [23] where the calibration method used is shown.

In order to use the models correctly, they developed a calibration algorithm to obtain the geometric parameters (*Kc, Kp, Rp* and $t_p$) and the colour parameters (*X* and *b*). To do this, through Equations (1) and (2) that define the geometric model and a homography *(H)*, they developed the warping functions, which relate a point of the image in the camera with a point of the projector and a point from the camera image $x_c$ with the point $x_s$ of the surface plane image [23].

$$w_p(x_c) = H_{pc}x_c, \tag{4}$$

$$w_s(x_c) = H_{sc}x_c. \tag{5}$$

Finally, the equations are entered into the Equation (3) to obtain the colour of the pixels at the camera point $x_c$, as shown in Equation (6).

$$p_c(x_c) = p_s(w_s(x_c))x[gXp_p(w_p(x_c)) + a] + b. \tag{6}$$

Once the calibration that provides us with the necessary parameters of the colour and geometric models was developed, they developed a cost function and its minimization function to optimize the system towards correct alignment [23].

Wang et al. (2010) established the alignment between the camera and the projector through a beam splitter, which allowed the light signal to be divided so that the same signal was perceived by the projector and the camera regardless of the detection distance. By performing this synchronization between the camera and the projector, a feedback effect was produced since the projected signal was part of the scene to be detected by the camera sensor. To avoid this phenomenon, the projector emitted in the visible range and the camera detector only detected IR wavelengths, thus preventing the virtual image from interfering with the projection of the following table [10].

Stenger et al. needed to develop a calibration system whereby the compensation image projected onto Mark Rothko's murals was displayed correctly. To do this, they first used the Matlab's control point selection tool to generate a geometric transformation and thus adjust the resolution of the target image and the current image of the artwork. Then, a lighting matrix with the levels of the three equal RGB channels was projected onto the work, in order to create a calibration curve for each channel. These calibration curves together with the colour mixing matrix, created to compensate the dependence of the channels on each other, created a suitable compensation image. In order for the compensation image to be positioned in the correct position relative to the illustration, a very irregular calibration image was projected onto the illustration and an image was captured. Then a Harris corner detector related corresponding points on the captured image and the calibration image. Using these points, a pattern was created in order to make the corresponding geometric transformation. Finally, by means of a RANSAC algorithm, outliers were eliminated, through an iterative non-linear fitting procedure [24].

Deformation Lamps tries to achieve the illusion of the perception of movements in objects by projecting an optimized light pattern on them. To do this, Kawabe et al. (2016) described an algorithm that consisted of defining a dynamic image sequence in order to project it onto a static object, thus achieving the perception of movement on the object.

To obtain this sequence of colour images, it was assumed that an image sequence, $I_{movie}$, can be calculated as a linear combination of a static colour picture $I_{static}$ and a dynamic grayscale image sequence $I_{luminancedynamic}$ [12].

$$I_{movie}(x,y,t) \approx I_{static}(x,y) + I_{luminancedynamic}(x,y,t), \tag{7}$$

Finally, by means of Equation (8), the dynamic luminance is calculated, which by means of Equation (9) is projected onto the image, $w$ being the factor that modulates the contrast of the dynamic component of the image and $B$ is an arbitrary gray background so as not to take values below 0.

$$I_{luminancedynamic}(x,y,t) = I_{luminanceseq}(x,y,t) - I_{luminancestatic}(x,y) \tag{8}$$

$$P(x,y,t) = wI_{luminancedynamic}(x,y,t) + B \tag{9}$$

For the deformation lamps method, Kawabe et al. used a manual alignment between the projector and the object, since for this method on 2D objects, no specific calibration is needed to obtain the visual effect they were looking for [12].

In Aleksić and Jovanović (2018), instead of using an algorithm yielding a real-time compensation image, a meticulous analysis of the artwork was carried out in order to be able to restore it digitally. This procedure was performed at the discretion of restoration professionals and then projected onto the work under controlled lighting conditions. For the alignment between the projected image and the actual artwork, the curvature of the image projection (curvilinear projection) as well as the perspective distortion were determined. The compensation images were projected at an angle and this perspective generated certain distortions of the image. In order for the compensation image to geometrically match the painting surface, the projection image curvature and perspective-generated distortion were determined so that the resulting image geometrically coincided with the surface of the artwork [20].

Vázquez et al. (2020) calibrated the projector to emit the corresponding lighting onto the work through a calibration algorithm and a merit function. Since the spectral power distribution of the projector (SPD) calculated is not the same as the real spectral emission of the projector ($D_{PK}$), an algorithm was developed that relates both distributions through the $Z$ factor. The purpose of the merit function is to minimize the colour difference ($\triangle E_{00}^*$) between the original and the restored artwork. The $\triangle E_{00}^*$ was calculated with CIEDE200. To eliminate distortions between the camera, the projector and the printed image, a $T$ transformation is developed through the Matlab image processing toolbox [21]. Finally, the method of correspondence between pairs of images of Vincent and Laganiere (2005) was used for the alignment between the projected image and the printed image [21,25].

## 4. Lighting

Cultural heritage objects must be exhibited for their appreciation, but inadequate temperature and lighting conditions can cause their deterioration [14,15,26]. Generating yellowing, discoloration or colour variation, corrosion, alteration and corruption, increase in surface temperature, and acceleration of deterioration on works of art [16]. This section reviews the main studies related to the lighting of artworks.

The damage caused by the emission of UV and IR radiation is controlled by filters or LED sources, but the radiation emitted by the visible range of the electromagnetic spectrum is necessary for the appreciation of the artwork and at the same time could produce irreparable damage to it [15]. The International Commission on Illumination (CIE) differentiates between the damage caused by the photochemical effect on different pigments and the damage due to the thermal action of light [26].

Photochemical damage is produced when a photon is absorbed by a pigment, causing the chemical properties of this pigment to change, manifested in artworks as changes in the mechanical properties of the pigments or in their colour [14]. The damage caused by thermal action, according to the CIE, has been more ignored in museums, since the damage has not been as visible as in the case of damage by photochemical action. Elena Lucchi (2016) evaluated the energy and environmental quality of museums, based on 50 European institutions. The result was that light is the most important environmental parameter in museums because it is directly related to the preventive conservation of works of art and to the comfort of viewers [27]. As the control of photochemical damage in museums has gained importance, so has the control of damage by the thermal action of light, taking into account that an increase in temperature favours chemical action at the molecular level in the different pigments [26,28–30]. Recently, a process to minimize the risk of deterioration in multifuncional historic buildings has been developed. It puts forward the profitability associated with the lighting parameters optimization in order to avoid damage to the cultural heritage shown in the exhibition [31].

The CIE made recommendation 157: 2004 to regulate the damage caused by photochemical action and the effects of thermal radiation of light on the different artworks exhibited in museums. Knowing that not all materials respond in the same way to the harmful effects of visible radiation, the CIE established, in its regulations, maximum irradiance limits and a defined exposure time to illuminate artworks, depending on the sensitivity to light radiation of the materials that constitute the artwork [26,28]. Mayorga et al. (2015) introduced the concept of Global Risk Factor (GRF), presenting a system for quantifying the damage produced by natural light, so that it could be used as the main or secondary source, minimizing the intrinsic risks it carries and using its advantages in lighting such as its ability to save energy, non-polluting, renewable source, psychological comfort, circadian cycle, and colour reproduction [32].

In addition to taking into account the damage due to radiation in museum exposures, it is important that the source used does not produce distortions in the perception of the artwork. For this, the colour rendering index (CRI) is usually used, which defines the colour reproduction capacity of a light source [26].

Even so, not all pigments used in an artwork respond in the same way to the effects of radiation, and not in a linear way [15]. This section reviews the main techniques developed for lighting in museums, which, in addition to protecting artworks from the effects of light, allow us to correctly appreciate them.

Historically, incandescent or fluorescent lighting was responsible for lighting in museums [15,33], however, in recent years, LED lighting has become very important for this application. LEDs offer great advantages, such as compactness, long useful life, adjustable intensity [33] and absence of radiation in the ultraviolet range and infrared range [33,34]. In addition, the radiation in the visible spectrum is reduced compared to continuous spectrum emission [34], their surface temperature rarely exceeds 50 °C, allowing them to be placed near wooden objects [14]. In recent years, LED technology has gained higher luminous efficiency, and by combining it with phosphors, it is easy to optimize the spectrum for the desired application [33].

Delgado et al. (2010) presented an optimization strategy applied to improve the luminous efficiency of a light source to illuminate artworks, excluding the part of the visible spectrum that does not contribute to the adequate perception of the illuminated object [35].

Berns et al. (2011) optimized a simulated triband light source made up of three LEDs with the fundamental objective of illuminating without causing damage to the artwork and producing the same colour rendering as the D65 illuminant, which is usually used as a reference in simulations due to its similarity to daylight. To this end, they performed four simulations using the emission of LEDs with bandwidths of 25 nm, 50 nm, 75 nm and 100 nm. The LEDs with bandwidths of 50, 75 and 100 nm provided emissions almost identical to D65, and the LED with a bandwidth of 25 nm had its efficiency improved by combining it with LEDs of greater bandwidth, obtaining very favourable results of $\triangle E$, colour reproduction and luminous efficacy, these being 1.2 $\triangle E_{94}^*$, 92 CRI, and 325 lm/W,

respectively. The 1.2 $\triangle E_{94}^*$ is imperceptible to the human eye in lighting conditions between 50 and 200 lux [34].

Furthermore, Berns (2011) developed an optimization algorithm capable of increasing or decreasing chroma depending on the demands of the artwork. On the one hand, an increase in chroma would make it possible to compensate for the lack of colour, respecting in any case the artist's intention. On the other hand, works exhibited in caves or churches must maintain their low chroma status since they were conceived that way [34].

Using an optimized white LED illumination obtained through the combination of different LEDs, Vienot et al. (2011) recreated the appearance of a bird specimen. They were able to intensify the saturation of faded colours by taking advantage of the distortions caused by narrow-band LED lighting [33].

Durmus and Davis (2015a) developed a lighting system with an optimized emission spectrum in order to produce energy savings without altering the appearance of the illuminated object. To do this, they started from the idea of dispensing with the emissions at those wavelengths absorbed by the object, designing a lighting system tuned by detectors with the reflectance of the objects, emitting fundamentally at the wavelengths reflected by the object and perceptible by the human eye. In this way, they managed to develop a lighting system through which an energy savings of 44% can be obtained and without appreciable changes for the human eye in the appearance of the object, with low values of $\triangle E_{ab}^*$ calculated in the CIE 1976 colour space L * a * b * [36].

Durmus and Davis (2015b) further perfected that technique by implementing added bandwidths of the order of 10 nm to the peaks of the selected emission spectrum, obtaining energy savings of between 55% and 71% maintaining low values of $\triangle E_{ab}^*$ [37].

Mayorga et al. (2016) analysed how lighting affects 23 different pigments in order to develop a system to measure the spectral response of exposed materials. This system allowed estimating the time and the maximum irradiance with which a artwork can be illuminated without noticeable changes in its colour [38].

Stenger (2015) based the lighting of Mark Rothko's murals on the limitation established by the CIE of an illuminance of less than 50 lux, restricting UV and IR radiation [26], since, in addition, one of the artist's requirements was that the artwork not be too illuminated [24].

Taking into account the need to develop a lighting system capable of displaying artworks without causing damage to them, the Zeus project emerged. It has been mainly promoted by scientists from the Complutense University of Madrid. This project also includes institutions involved in the conservation and dissemination of cultural heritage objecds and researchers in Communication and Audiovisual Technology. Among other strategies related to the analysis of artworks for their correct conservation and visualization, the Zeus project developed a selective lighting system that calculates the appropriate light distribution point by point based on the characteristics of the artwork [30].

Durmus et al. (2018) designed an optimized lighting system by which the reflectance of the artworks was obtained in order to be able to project onto them an illumination capable of reducing photochemical damage due to spectral absorption of radiation in the pigments used based on the reflectance spectrum of the pigments. In this development, it was essential not to affect the appearance of the objects so as not to alter the intention of the artist. To this end, they used multi-objective genetic optimization algorithms (MOGA). Genetic algorithms (GAs) are based on the theory of evolution, where, starting from a wide range of species, only those suitable survive, the rest facing extinction. The MOGAs, unlike the GA, allow a greater number of solutions by defining an aptitude function that evaluates the solutions at each step, thus generating more precise solutions. Considering that when using a visible spectrum light source, visibility and damage conflict, they used MOGA optimization algorithms since by themselves they can produce more than one result that does not allow the optimal solution. Therefore, it was essential to use Pareto's optimal solution technique, which was responsible for selecting alternatives when there is no single optimal solution, allowing obtaining an optimized lighting source to minimize absorption damage without causing a notable damage to the colour of the artwork [29].

Carla Balocco and Giulia Volante set out a methodology for sustainable cultural heritage lighting that not only complied with the regulations for the correct conservation and protection of works of art, but also provided visual and perception comfort from the point of view of observer reducing the power consumption of the light source. Thus, they carried out a study to illuminate those areas of interest of the work of art by analyzing the eye movements of a series of subjects when viewing it [39].

Muñoz De Luna, in his doctoral thesis published in 2017, developed a point-by-point spectral reflectance measurement system using non-contact spectrophotometers in order to subsequently carry out a colourmetric analysis of the artworks that allows a temporal monitoring of these in an objective. This system was applied to study the following works of Pablo Picasso: *Guernica* and *Woman in Blue*. Using Michiel Sweert's *Boy with a Turban and Corsage*, he carried out a comparative study of the colourimetric values of the artwork, illuminating it using the illuminants that the CIE defines as standard illuminant, finding in certain areas of the work, appreciable colour differences for human perception. Finally, using the reflectance measurement technique previously described and the knowledge of how the different illuminants affect the conservation and appreciation of an artwork, there was studied the lighting used for the cave paintings of the cave of *El Castillo* in Puente Viesgo (Cantabria). To this end, an LED lighting system was developed using a minimum optimization algorithm that allowed the emission of an optimal spectral distribution in terms of minimal deterioration, maximum contrast between the colour stimulus and the background, and minimum chromatic difference between the observed painting under the light used by the artist and observed with the developed illuminant [28].

The lighting system developed by Vázquez et al. (2020) for projecting onto artworks was optimized to minimize radiation damage, while also producing a restored appearance of the art [21]. With the information from the CIELab coordinates and reflectance information, they developed a merit function based on Fernandez-Balbuena et al. [40] to optimize the $\triangle E_{00}^*$ between the D65 reference and test illuminant. A second merit function was developed in order to minimize the radiation damage through the concept of GRF, developed by Mayorga et al. [32,38]. Using the optimization algorithm, based on Durmus and Davis [36,37] and Muñoz De Luna [28] described above, and the lighting system described by the Zeus Project [30], they calculated the optimal SPD and the intensity of each point in the image [21].

Rui Dang et al. proposed a method of quantifying damage taking the type of paint, the level of illuminance and the exposure time as three variable units, thus allowing to optimize lighting systems in a much more precise way. They found that some works of art allow higher levels of illuminance than those set by the regulations, thus achieving an improved visualization by the viewer [41].

## 5. Conclusions

This review article has treated the main studies that have been carried out in recent years that use the projection mapping technique. Its purpose has been to provide the necessary help related to the calibration and lighting processes that may be required in future applications.

In the section on calibration techniques, different viable methods have been described to achieve the alignment between camera and projector, such as the structured light method [10,22], the beam splitter method [10], and the use of the Harris corner detector [24]. Furthermore, in many of the studies, different limitations have been taken into account, such as feedback, or the minimization of $\triangle E$ in virtual restoration applications, proposing in both cases suitable solutions.

When using the projection mapping technique, attention must be paid to adequate lighting conditions, especially in those applications related to the exhibition and conservation of cultural heritage objects. For this reason, in the fourth section of this article, there have been reviewed the main lighting optimization studies whose aim is to minimize the change in appearance of the illuminated surface while reducing the harmful effects of radiation.

**Author Contributions:** Conceptualization, Á.G.M; Methodology, Á.G.M. and J.C.M.A.; Validation, Á.G.M, J.C.M.A. and A.J.B., Formal Analysis, Á.G.M; Investigation, Á.G.M; Data Curation, Á.G.M; Writing—Original Draft Preparation, Á.G.M; Writing—Review & Editing, J.C.M.A. and A.J.B.; Visualization, Á.G.M; Supervision,

Á.G.M.; Project Administration, Á.G.M.; Funding Acquisition, Á.G.M. All authors have read and agreed to the published version of the manuscript.

**Funding:** This work has been funded by project number RTI2018-097633-A-I00 of the Ministry of Science and Innovation of Spain, entitled 'Photonic restoration applied to cultural heritage: Application to Dali's painting: *Two Figures*.'

**Conflicts of Interest:** The authors declare no conflict of interest. The funders had no role in the design of the study; in the collection, analyses, or interpretation of data; in the writing of the manuscript, or in the decision to publish the results.

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
