# Peer review of "Virtual Restoration and Visualization Changes through Light: A Review"

_heritage, doi:10.3390/heritage3040076_

Round 1
Reviewer 1 Report
The paper presents a review on the virtual modification of the appearance of an heritage object using lighting technologies. The topic is really new and interesting. The introduction unfortunately is not clear for the lesole that are not working on it. I suggest you to define better what is a argumented reality and virtuality, referring to standard, papers and examples. Also a focus on heritage object is needed. The sim and the methodology of the paper are not clear. You refer also to the damage problems related to light in artworks (changes of their appearance due to the action of external agents, such as humidity / temperature / electromagnetic radiation). On this topic there is big literature real used from 1970 untile now. A preventive conservation program must begin with an intensive museum-wide study which examines the collection, the indoor environment, the building, and the lighting and hvac system. The collection should be examined since different materials require different handling and environmental conditions. I understand that it is not possibile inserting all the papers, but in a review paper it is necessary to insert the most famous one (I.e. The first book of Thompson on the museum environment, the most recent book of Camuffo on microclimate on cultural heritage, and a comprensive review Paper on preventive conservation in museum buildings) These 3 documents consider the impact of non correct indoor parameters on heritage object and the Relationship among them (temperature, humidity, light and contaminants) on it. Particularly, in the last one, you can find an overview on old and new trends in this discipline. Some papers on this topics were collected and commented. I suggest to refer to them, as you semplice to much the problem (from row 66). Also, in the part in lighting you tested only the light damage. The part on calibration is well detailed and referenced. In the part on lighting you wrote: “The damage caused by thermal action, according to the CIE, has been more ignored in museums, since the damage has not been as visible as in the case of damage by photochemical action”. I agree with you, thermal aspects are not considered. In many museum there is a thermal control (with hvac or thermal controlled exhibits) thus the reduction of thermal damage is supposed connected with them. the paper https://doi.org/10.1016/j.enbuild.2016.02.037 compare several European museum obtaining your result: light is the most considered environmental parameter, relative humidity is few considered as it caused an irreversible damage, temperature and air quality are not considered at all). Thus you can refer to this example to emphasize you idea on the importance of light. As the control of photochemical damage in museums has gained importance during the years, mani paper deal with this. For example, https://doi.org/10.3390/su10051671 investigated the physical connection between light, perception, and information. The paper https://doi.org/10.3390/su12125155 demonstrates that the interventions on lighting systems are cost-efficiency solutions and low-engineering interventions, that improve the life expectancy of previous museum objects. Finally, https://doi.org/10.1016/j.buildenv.2020.107142 presents a methodology to evaluate the lighting in museum, proposing correlated color temperature of LEDs for highly light sensitive artworks. I suggest you these very recent references to improve and updated your scientific review in a more comprehensive prospective on preventive conservation and risk assessments in museum objects. Conclusion are not well discussed, and the discussion using different bullet points permits to focus on the most important results. English is fine.
Reviewer 2 Report
A well-written paper with a concise overview of the technology. This offers the general reader an understanding of the importance of light projection as a technique to alter, or enhance, the display of museum objects.
Reviewer 3 Report
The present manuscript provides a review of virtual restoration of cultural heritage objects through the use of projection mapping techniques.
In general, I think that the topic is definitely interesting for Heritage Science readers since it provides an overview of how cultural heritage objects can be better fruited through digital restoration.
Nonetheless, the manuscript suffers from the fact that it consists of text only. In this sense, I strongly suggest authors to add images displaying the results achievable through digital restoration. I think that this addition would really help readers to immediately understand the results achievable through digital restoration without necessarily reading the quoted references. Proper images could be made available by quoted authors or could be made available from already published papers provided that copyright issues will be managed.
Further, the manuscript is somehow chaotic in some parts. To improve readability and comprehension I suggest authors:
1) to add a scheme to better clarify the different fields of application of digital restoration;
2) to add a scheme to better illustrate the different calibration methods.
In the following I detail better these suggestions together with other further comments:
- Lines 66-68: references 14 and 15 are not specifically focus on degradation of artworks induced by external agents or artificial/natural illumination. Authors should find proper references for this topic.
- Section 2: Projection mapping applications: as previously quoted, here some figures illustrating the use of virtual restoration would highly improve the quality of the paper;
- Section 2: Projection mapping applications: I further suggest to add a scheme to illustrate the different use of virtual restoration (as: a) the virtual removal of the varnish layer on a painting; b) the virtual restoration of a faded painting; c) the virtual projection of the original color/manufacture on a sculpture; and so on).
- Lines 90-98: these lines describe the layout of the paper. They should be moved at the end of the introduction, where the layout of the entire paper should be provided.
- Section 3: Calibration: in general this section is hard to be understood by non-experts in the field of virtual reality (as I suppose are most of the readers of Heritage Science). In order to improve readability and comprehension, I first suggest to provide a scheme for differentiating the different calibration methods (i.e. based on structured light, beam splitters and Harris corner detector). Further, I’m not really sure that the quoted equations (1 to 9) would be understood by Heritage Science readers. I suggest again to try to provide a scheme for each type of calibration method to explain its working principle.
- Section 4: Lighting: In my opinion this section actually does not deal with virtual restoration or with visualization changes, but it deals with the correct illumination of an artwork in order not to damage it with excessive electromagnetic or thermal effects. Hence, while being very interesting, this section is out of topic. I suggest to revise the title and the abstract of their paper in order to include this different topic, too.
Round 2
Reviewer 1 Report
-